# The Timeliness of Drug Therapy in Colorectal and Prostate Cancer in Antigua and Barbuda: The Role of Disease Stage

**DOI:** 10.3390/healthcare13080915

**Published:** 2025-04-16

**Authors:** Andre A. N. Bovell, Jabulani Ncayiyana, Themba G. Ginindza

**Affiliations:** 1Discipline of Public Health Medicine, School of Nursing and Public Health, University of KwaZulu-Natal, Durban 4000, South Africa; ncayiyanaj@ukzn.ac.za (J.N.); ginindza@ukzn.ac.za (T.G.G.); 2Cancer & Infectious Diseases Epidemiology Research Unit (CIDERU), College of Health Sciences, University of KwaZulu-Natal, Durban 4000, South Africa

**Keywords:** Antigua and Barbuda, stage, colorectal cancer, prostate cancer, time interval, predictor, drug therapy, regression analysis

## Abstract

**Background/Objectives**: Colorectal and prostate cancers are significant public health problems for countries globally. In Antigua and Barbuda, where resources are limited, there is a need for both insight and evidence on the timeliness of drug therapy initiation for colorectal and prostate cancers as a way of improving disease management capabilities and prognostic outcomes for diagnosed cases. This study aimed to investigate whether the disease stage of colorectal and prostate cancers is a predictor of the time to drug therapy initiation for persons diagnosed with these cancers in Antigua and Barbuda from 2017 to 2021. **Methods**: This was a retrospective analytical study that utilized data, inclusive of the coronavirus disease 2019 effect, for colorectal and prostate cancer patients extracted from four study sites in Antigua and Barbuda to assess the relationship between disease stage and time to drug therapy initiation. Analyses were performed using polytomous multivariable logistic regression modelling. **Results**: Analyses showed that the final models for both cancers were significant (*p* < 0.05); however, disease stage was not a predictor of time to drug therapy initiation in either model. The ORs observed were 41.58 (95% CI: 0.78–2219.28) for colorectal cancer and 0.41 (95% CI: 0.11–1.44) for prostate cancer. **Conclusions**: Regarding both cancers, our findings demonstrate that disease stage alone is not a significant predictor of time to drug therapy initiation unless analysed alongside other essential patient characteristics in each respective model. Our findings are a useful reference that can be utilized by policymakers to improve treatment capabilities, including establishing a standardized care algorithm to optimize timeliness in administering drug treatment for these cancers in Antigua and Barbuda.

## 1. Introduction

Cancer burden is a significant public health challenge in every country [1,2]. Globally, there were approximately 20 million new cancer cases and around 9.7 million cancer-related deaths in 2022 [1], with projected increases in several countries. For example, estimates from the United States suggest that this nation will record more than 2 million new cancer cases and 618,120 cancer deaths in 2025 [2].

Across several countries, prostate cancer ranks as the most frequently diagnosed cancer in men and the second cause of cancer-related mortality among men [1], while colorectal cancer has become the second and third most common cancer in men and women, respectively [1]. In 2022, prostate cancer was responsible for 7.3% of all new cancers diagnosed worldwide [1] compared with colorectal cancer, which was responsible for almost 9.6% of cancers diagnosed [1].

Globally, men of African ancestry in the Caribbean have some of the highest incidence rates of prostate cancer [3], with elevated cumulative risks of prostate cancer deaths observed in countries such as Trinidad and Tobago, Guyana, and Barbados [4].

While colorectal cancer is associated with a diverse number of risk factors, and the chance of developing this cancer increases markedly after 50 years of age, prostate cancer is known to have established risk factors, such as advancing age, family history, and certain genetic mutations, and it still has an unknown aetiology [3].

For colorectal cancer, surgery is the only possible cure for the removal of tumours, with the majority of patients undergoing surgery [5]. Additionally, postoperative chemotherapy may be required in instances where the primary lesion has spread to the lymph nodes [6]. For rectal cancers, usually, preoperative radiotherapy in combination with chemotherapy is required [6]. Where there is generalized disease, chemotherapy is the preferred treatment, with surgery being used for metastatic removal [6]. Prognosis strongly depends on the stage at diagnosis, and the disease can mostly be cured if diagnosed at an early stage [7]. Adjuvant chemotherapy (AC) has been shown to improve outcomes for many patients at different clinicopathological stages of disease [8], with studies suggesting heterogeneity in response to drug therapy, even within the same pathological stage [9].

Regarding prostate cancer, and because of its variable morphology, clinical behaviour, and prognosis, the treatment of choice for patients includes the use of androgen deprivation therapy (ADT) in some clinical settings (neoadjuvant, concurrent, and adjuvant) with or without the use of radiation [10,11,12,13], even though its use is not curative in patients with locally advanced or metastatic disease [11,13]. In effect, treatment consists of a luteinizing hormone-releasing hormone (LHRH) agonist, an LHRH agonist with a first-generation antiandrogen, or an LHRH antagonist, i.e., agents that are normally administered continuously [11].

In Antigua and Barbuda, where colorectal cancer was ranked as the third leading cause of cancer deaths and the tenth leading cause of death overall from 2000 to 2019 [14], as well as where prostate cancer accounted for 45% of all cancers in men during the period 2001–2005 [15], understanding how certain health determinants can impact the burden of these cancers in the population holds tremendous value, including disease prognosis and managing other outcomes [11,16].

Currently, there is no published evidence in the country on the extent of the relationship between the disease stage of these cancers and the initiation of drug therapy in patients on the island; that is, time to drug therapy initiation and disease stage as a way of assessing cancer burden has not been explored.

On the basis that the systematic collection of data on colorectal and prostate cancers will help provide both objective insights and measurable evidence of Antigua and Barbuda’s colorectal and prostate cancer management capabilities [16], this study aimed to assess whether disease stage is a predictor of time to drug therapy initiation in colorectal and prostate cancer cases diagnosed in Antigua and Barbuda between 2017 and 2021.

## 2. Materials and Methods

### 2.1. Study Design, Setting, and Population

We used a retrospective analytical study design that used some of the data and results previously reported in the published article “Incidence, trends and patterns of female breast, cervical, colorectal and prostate cancers in Antigua and Barbuda, 2017–2021: a retrospective study” [17] to investigate the relationship between disease stage and time to drug therapy initiation for patients with colorectal and prostate cancers in Antigua and Barbuda.

We used data from patients who were >18 years old and had a diagnosis of a primary malignancy of the colorectum and prostate between 1 January 2017 and 31 December 2021 [17]. Patients with recurrent disease were not included in the study [17].

### 2.2. Sample Size

Consistent with the study’s use of data reported in a previous publication [17], the study’s objective and because of convenience, we sought to utilize data for all cases diagnosed with colorectal (n = 79) and prostate cancer (n = 109).

### 2.3. Data Collection and Management

As mentioned in an earlier study [17], the data used in this study were extracted from patient records of persons diagnosed—as per the International Classification of Diseases Tenth Edition (ICD-10) ((C18, C19, C20-colon/rectum) and (C61-prostate)) [18] and archived with either the Oncology, Pathology and Urology departments of the Sir Lester Bird Medical Centre (SLBMC) and The Cancer Centre Eastern Caribbean, Antigua and Barbuda (TCCEC)—with colorectal or prostate cancers [17]. We utilized a two-stage process to obtain patient data [17]. In the first phase, we used a predesigned and pretested data collection form to abstract text-based, cancer-specific, and patient-related demographic and clinicopathological information from the previously identified departments of the Sir Lester Bird Medical Centre and electronic-based data of a similar nature from The Cancer Centre Eastern Caribbean [17]. Additional text-based data on cancer cases, including demographic, drug treatment, and socioeconomic information, were obtained from the Medical Benefits Scheme [17]. Electronic-based data on cancer deaths were collected from the Health Information Division, Ministry of Health, Antigua and Barbuda [17]. In the second phase, we cross-referenced the abstracted records using a combination of unique identifiers [17,19]. For cases collected from the Medical Benefits Scheme (MBS), we used the Medical Benefits Scheme identification number (MBS number) [19]. For records collected from the SLBMC, we utilized the hospital-generated medical patient identifier (MPI) and the MBS number. For cases from TCCEC, we relied on the unique TCCEC number and the MBS number. Concerning data on cancer deaths, we utilized the MPI and/or MBS number [19]. This approach helped to eliminate duplicate records and improve the completeness of the data collected [19].

The data were de-identified and anonymized at the time of collection [17]. To assist with analyses, we categorized our data as follows: demographics, including age as both a continuous variable and grouped in stated categories of <40, 40–50, 50–60, 60–70, and >70 years (note 40–50 means, 40 and up to but not including 50 years, etc.) for both cancers, parish, or area of residence, a dichotomous variable (all other parishes—[Barbuda, St. George, St. Peter, St. Mary, St. Phillip, St. Paul], and St. Johns); year of presentation (2017 to 2021); vital status (alive and dead); clinicopathological, including the coronavirus disease effect and evidence of noncommunicable diseases other than cancer (yes and no); treatment, including radiation therapy status (incomplete/unknown and completed/known); and socioeconomic status, including employment status at presentation (employed and not employed). Additional characteristics are outlined in Table 1 and presented in the Results Section of this article.

The coronavirus disease (COVID-19) effect, a categorical variable, was divided into two groups: (i) the pre-COVID-19 period (presentation period from 1 January 2017 to 28 February 2020); and (ii) the COVID-19 period (1 March 2020 to 31 December 2021). This was based on Antigua and Barbuda’s recording of its first coronavirus disease (COVID-19) patient on the 13 March 2020 [20].

### 2.4. Outcomes Ascertained

The primary outcome determined was whether disease stage was a predictor of time to drug therapy initiation for cases of colorectal and prostate cancers in 2017–2021. Given the differences in data capture, the key exposure, outcome variables, and available covariates by cancer type were not entirely similar.

Exposures: The main exposure was disease stage at diagnosis, a dichotomous variable with the groups ‘Early-stage disease (clinical stages 1 and 2)’ and ‘Late-stage disease (clinical stages 3 and 4)’.

Outcome: The main outcome was the time to systemic drug therapy initiation, a categorical variable that was divided into three (3) groups: (i) no drug therapy in the period, (ii) early time (≤3 months), and (iii) late time (>3 months).

Time to drug therapy initiation was defined as the time, in months, from the date of the initiation of drug therapy (systemic therapy for colorectal cancer and androgen deprivation therapy–luteinizing hormone-releasing hormone agonists for prostate cancer) back to the date of diagnosis for each patient.

For colorectal cancer, covariates assessed as possible predictors of time to drug therapy initiation included age at presentation, the coronavirus disease effect, employment status at presentation, histological grade, parish (area of residence), lymph nodal status, distant metastases, and the number of care-related payments made to the hospital [1,21].

For prostate cancer, the covariates assessed as possible predictors of time to drug therapy initiation included age, the coronavirus disease effect, Gleason score, noncommunicable diseases other than cancer, parish (area of residence), radiation therapy status, the number of care-related payments made to the hospital, and employment status at presentation [13].

### 2.5. Data Analysis

Colorectal cancer analysis considered (i) demographic variables, such as age (continuous variable), age at presentation (categorical variable), area of residence (parish), and COVID-19 effect; (ii) clinicopathological variables, which included lymph nodal status, disease stage, histological grade of disease, distant metastases status, tumour dimensions, tumour site, and noncommunicable diseases other than cancer; (iii) socioeconomic variables, such as employment status at presentation; and (iv) treatment variables, such as chemotherapy treatment.

Prostate cancer analysis considered (i) demographic variables, such as age (continuous variable), age at presentation (categorical variable), area of residence (parish), and COVID-19 effect; (ii) clinicopathological variables, which included disease stage, Gleason score, and noncommunicable diseases other than cancer; and (iii) socioeconomic variables, such as employment status at presentation. Other variables are noted in Table 1 of the Results Section of this paper.

For both cancers, descriptive analysis was conducted to gain a comprehensive understanding of the characteristics of the respective study populations, the data, and the distribution of variables [22]. Some variables were considered a priori confounders (age at diagnosis) and effect modifiers.

For both cancers, univariable analysis was used to describe the epidemiological characteristics of the cancer patients, and they involved summarizing variables, which used frequency tables and assessing the frequency distribution of these variables [22,23]. Bivariable analysis via cross-tabulations of each exposure variable against the main outcome variable was performed using the chi-square test or Fisher’s exact test, where appropriate, to produce estimates of the percentages and frequencies of the investigated variables [22].

Polytomous (multinomial) logistic regression was performed using the main outcome against the primary exposure to derive crude odds ratios (ORs) and corresponding 95% confidence intervals (CIs). Its selection was made given its advantage of effectively handling outcome variables with two or more categories [24]. Additionally, its utilization was premised on its ability to preserve essential data information, thus making it easier to manage our data and leading to a better interpretation of the eventual study results [24].

Polytomous (multinomial) logistic regression was repeated for the other covariates against the categories of the outcome variable. Crude or unadjusted measures of association (OR) were estimated. Confidence intervals (95%) were computed and chi-square tests for categorical data were used in identifying variables that have attained statistical significance at a *p*-value ≤ 0.05. Logistic regression was used to assess whether any covariate confounded the relationship between disease stage and time to drug therapy initiation, and a likelihood ratio test was performed to ascertain the evidence of an association between the main exposure and each covariate at the 5% level, with *p* ≤ 0.05 taken as evidence of an association.

Polytomous (multinomial) logistic regression was subsequently used for computing adjusted measures of association (ORs), the corresponding 95% CIs and *p*-values from the significance test between the outcome time to drug therapy initiation and key exposure disease stage at diagnosis, and each covariate of interest in turn. The likelihood ratio statistic (LRT) was used to assess the contribution of each variable to the model with the main exposure and outcome variables. Variables that either resulted in statistically significant odds ratios (ORs) (*p*-value ≤ 0.05) or marginally statistically significant odds ratio (ORs) (*p*-value < 0.10) to cater for probable within sample variations [25,26], or were found to be clinically relevant based on the literature, even if not found to be statistically significant, were considered for inclusion in the fully adjusted multivariable regression model [27]. For colorectal cancer, these were the following: (i) clinical relevance based on the literature—age, vital status, disease stage, histological grade, noncommunicable diseases other than cancer, chemotherapy treatment, and year of presentation [28], and (ii) statistical significance—vital status, disease stage, area of residence (parish), and employment status at presentation.

For prostate cancer, these were the following: (i) clinical relevance based on the literature—age (categorical variable), vital status, histological grade, estimated monthly income level at presentation, noncommunicable diseases other than cancer at presentation, and year of presentation [28].

Using the model with the outcome time to drug therapy initiation and main exposure disease stage at diagnosis, multivariable polytomous logistic regression was used to determine whether the relationship between these variables was affected by effect modification of the baseline age (reduced to a dichotomous variable with two categories, ≤50 years and >50 years). A value of *p* < 0.10 for the likelihood ratio test (LRT) was taken as evidence of effect modification, with any resultant interaction variable to be selected for the final model.

Multivariable polytomous logistic regression was used to fit the final model. This involved fitting the respective models with their corresponding covariates, following which each variable was omitted in turn from the models and their resulting *p*-values from corresponding LRT tests were noted [29]. The models with all chosen covariates included, except the ones with the highest *p*-value (>0.05) from the LRT, were run as before. This process was repeated until the *p*-values for each iteration of the LRT were assessed. Predictors that, through the above process, were omitted were added to the models in succession if their inclusion contributed to the model’s overall *p*-value being ≤0.05. The models depicting the best subset of variables with an overall *p*-value ≤ 0.05 were selected as our final models. An LRT was conducted on the final models to assess the null hypothesis of no association between the disease stages adjusted for the covariates in the model. All of the analyses were conducted using Microsoft Excel version 2501 and the STATA 17/SE-Standard Edition (Statistical Corporation, College Station, TX, USA) statistical package.

### 2.6. Ethical Considerations

Approval for this study was granted by the Antigua and Barbuda Institutional Review Board, Ministry of Health (AL-04/052022-ANUIRB), the Institutional Review Board of the Sir Lester Bird Medical Centre, and the University of KwaZulu-Natal Biomedical Research Ethics Committee (BREC/00004531/2022). This study did not involve direct contact with cases, and there was no direct risk to persons [30]. De-identification and anonymization were achieved by not documenting the names of the patients [30].

## 3. Results

### 3.1. Descriptive Statistics/Univariate Analysis

Colorectal Cancer—A total of 79 patients were initially considered for this study. This was reduced to 70 by listwise deletion to enable the use of complete information. For the 70 patients, the mean age was 66.3 (±11.72) years, with approximately 49% of them between the ages of 50 and 70 years (Table 1). Compared with other parishes (rural areas), 53% resided in St. John’s (urban area). The number of cases diagnosed each year in the five-year study period varied from 14% and 10% in 2017 and 2019, respectively, to 19%, 36%, and 20% in 2018, 2020, and 2021, respectively, with females accounting for 53% of all cases. Approximately 41% had early-stage disease, while 59% had late-stage disease. Grade 2 disease (moderately differentiated) at 83% was the dominant histological grade. Roughly 91% of cases had their tumour site determined, 60% had some form of chemotherapy treatment, 34% had an undetermined tumour dimension status, 59% had their distant metastatic status determined, 90% had their regional lymph node and primary tumour statuses determined, 73% had a stated tumour extent, 59% had a noncommunicable disease other than cancer, 49% had made upwards of 10 care-related payments to the main hospital, and 56% were employed at the time of presentation. Regarding the time to drug therapy initiation, 47% had no drug therapy administered, while 36% had therapy initiated at ≤3 months (≤90 days), and 17% had therapy initiated in the period of >3 months (>90 days) (Table 1).

Prostate Cancer—A total of 109 cases were considered for this study. The mean age of cases was 66.5 (±7.77) years [17]. Most of the patients were between the ages of 60 and 70 years (49%). Approximately 62% of cases resided in St. John’s parish, the year 2020 saw 37% of all cases, and 55% of cases occurred during the COVID-19 pandemic (2020 and 2021). Roughly 12% of patients died within the study period of 2017–2021, while around 54% of patients had early-stage disease and 46% had late-stage disease. Approximately 56% of patients had a Gleason score of either 6 or 7, 79% had an undetermined metastatic status, 73% had an undetermined regional lymph node status, and 72% had an undetermined primary tumour status. About 70% of cases had androgen deprivation therapy in the period, 51% had completed or known radiation therapy status, 39% had evidence of a noncommunicable disease other than cancer, 50% had ≤10 payments made to the main hospital, and around 26% were not employed at the time of presentation (Table 1). Regarding time to drug therapy initiation, 64% had no drug therapy administered, while 12% had therapy initiated in ≤3 months (90 days), and 24% had therapy initiated beyond 3 months.

### 3.2. Bivariable Analyses

Colorectal Cancer—Table 2A shows the crude estimates of the relationship between each exposure variable and the outcome variable time to systemic drug therapy initiation (Table 2A). Except for the variables chemotherapy treatment and number of care-related payments, with *p*-values < 0.05, no other variables appeared to be definitively associated with disease stage. The variables of tumour extent and employment status at presentation both appeared to be marginally associated with the time to systemic drug therapy initiation (*p*-value = 0.06). Assessing the relationship between the main exposure disease stage and each of the other covariates in turn showed that the variables parish, distant metastases, stated tumour extent, and evidence of noncommunicable disease other than cancer were the only variables that appeared to be associated with the main exposure (*p* < 0.05).

Examining the relationships between exposure disease stage and each covariate in turn, followed by an examination of the relationship between the outcome time to systemic drug therapy initiation and each of the other covariates in turn, showed that the variable ‘stated tumour extent’ was found to be associated with the exposure and marginally associated with the outcome variable, and it was therefore considered a possible confounder.

In the bivariate model with only disease stage and time to systemic drug therapy initiation, the results were nonsignificant (OR 1.25, 95% CI: 0.44–3.58, *p* = 0.68, OR 1.67, and 95% CI: 0.42–6.64; overall *p*-value = 0.75). After controlling for each covariate in turn, the ORs fluctuated noticeably, that is, marginally and/or markedly, with the variables chemotherapy treatment and number of care-related payments showing significance in the bivariable models (*p* < 0.01). No other variables showed significance in their respective bivariable models (*p* > 0.05) (Table 2A).

Additionally, age at presentation (as a dichotomous variable) was not an effect modifier of the relationship between time to drug therapy initiation and disease stage (LRT *p*-value = 0.59). We, therefore, assumed that the data were not affected by interaction and that the interaction term would be excluded from our final model.

Prostate Cancer—Table 2B shows the crude relationship between each exposure variable and the outcome time to androgen deprivation therapy initiation (Table 2B). Based on the analysis, except for the variables of distant metastases, regional lymph node status, and primary tumour, none of the other exposure variables, including the main exposure disease stage, were found to be crudely associated with time to androgen deprivation therapy initiation (*p* > 0.05). Assessing the relationship between the main exposure disease stage and each of the other covariates in turn showed that the variables age group and androgen therapy showed a significant relationship with disease stage (*p* <0.05). None of the other variables showed significance. The results, therefore, were suggestive of no evidence of a possible confounding factor affecting the relationship between disease stage and time of drug therapy initiation.

In the bivariate model, with only time to drug therapy initiation and disease stage, the results were nonsignificant (OR 0.91, 95% CI: 0.28–2.97; OR 0.66, 95% CI: 0.26–1.66; overall *p*-value = 0.67).

After controlling for each covariate in turn, there was a general fluctuation in the odds ratio of time to drug therapy initiation for most variables; however, except for distant metastases, regional lymph node status, and primary tumour (overall *p*-values < 0.05), none of the other variables showed significance in their respective models (*p* > 0.05) (Table 2B). No evidence of interaction was observed in the bivariate model comprising time to drug therapy initiation and disease stage when assessed for possible interaction by age group (*p* > 0.05). We, therefore, assumed that the data were not affected by interaction and that the interaction term would be excluded from our final model.

### 3.3. Multivariable Analyses

Colorectal cancer—Table 3A shows the results of performing polytomous multivariate logistic regression analysis by retaining, in the final model, variables based on statistical significance and clinical relevance (Table 3A). The variables chosen based on statistical significance were chemotherapy treatment and number of care-related payments; and variables selected based on clinical relevance were age group, parish, COVID-19 effect, vital status, disease stage, stated tumour extent, distant metastases, regional lymph node status, primary tumour, evidence of noncommunicable diseases other than cancer, and employment status at presentation. The variable histological grade was not included in the multivariable analysis because of data convergence issues. Following several iterations of multivariable logistic regression (using stepwise regression), the final model was determined as comprising the variables of disease stage, parish, year of presentation, vital status, sex, chemotherapy treatment, tumour dimension status, distant metastases, stated tumour extent, evidence of noncommunicable diseases other than cancer, number of care-related payments, and employment status at presentation. Year of presentation, chemotherapy treatment, stated tumour extent, number of care-related payments, and employment status at presentation showed significant results (Table 3A), with disease stage not being found as a significant predictor of time to drug therapy initiation in the model (Table 3A).

When compared with those who had not been administered drugs, there was, therefore, a higher risk of a late time to drug therapy initiation in persons with a year of presentation in 2018 and 2020 (OR 2816.41, 95% CI: 4.07-1948183; OR 2478.28, 95% CI: 3.43-1790327). The highest risk of early time to drug therapy initiation was found for persons who had some chemotherapy, and a lower risk of late time to drug therapy initiation was found for those who had chemotherapy but time to drug therapy initiation was more than 90 days. The highest risk of low time to drug therapy initiation was found for persons with a stated tumour extent but a time interval of 90 days or less; the highest risk of low time to drug therapy initiation was found for persons with >20 care-related payments; a much higher risk was found for persons who had care-related payments of >20 and a time to drug therapy initiation of >90 days; and a high risk of low time to drug therapy initiation was found for persons whose employment status at presentation was employed (Table 3A). Even though, overall, the model showed significant results (*p*-value < 0.001), no significant difference between the categories of time to drug therapy initiation was found among the other variables retained in the model (Table 3A).

The results of a likelihood ratio test (LRT) to assess the significance of disease stage adjusted for the other covariates showed that disease stage was not highly significant in the model (LRT = 4.44, *p* = 0.109). Additionally, a likelihood ratio test (LRT) that was conducted to evaluate the significance of the other covariates in the final model adjusted for disease stage showed that the other covariates were still highly significant in the model (LRT = 77.97, *p* < 0.001).

Prostate cancer—Table 3B shows the results of performing polytomous multivariate logistic regression analysis by entering in the final model variables based solely on their statistical significance and clinical relevance (Table 3B).

Variables chosen based on statistical significance were distant metastases and primary tumour; and variables selected based on clinical relevance were age group, parish, COVID-19 effect, vital status, disease stage, Gleason score, regional lymph node status, androgen deprivation therapy, radiation therapy status, number of care-related payments, evidence of noncommunicable disease other than cancer, and employment status at presentation. Following several iterations of multivariable logistic regression (using stepwise regression), the final model was determined as comprising the variables of disease stage, age group (years), Gleason score, distant metastases, lymph node status, androgen deprivation therapy, number of care-related payments, evidence of noncommunicable diseases other than cancer, and employment status at presentation. Only the variable of distant metastases showed significant results in the final model (Table 3B). Generally, disease stage did not appear to be a predictor of time to androgen deprivation therapy initiation (OR 0.19, 95% CI: 0.03–1.13; OR 0.41 (0.11–1.44)) (Table 3B). Similar observations were made for all other remaining covariates in the final model. A higher risk of early time to androgen deprivation therapy initiation was evident for persons whose metastatic status was determined and whose time to drug therapy initiation was < 3 months; similar observations were made for persons who had androgen deprivation therapy compared with those who had no record of androgen therapy stated (Table 3B). Even though the model showed significant results (*p*-value = 0.019), no significant difference between no therapy administered and either early-time or late-time androgen deprivation therapy initiation were found among the other variables retained in the model (Table 3B).

Performing a likelihood ratio test to assess the significance of disease stage adjusted for the other covariates showed that disease stage was not significant in the model (LRT = 4.50, *p* = 0.105). Additionally, a likelihood test performed to test the significance of the other covariates in the final model adjusted for disease stage showed that the other covariates were still highly significant in the model (LRT = 39.69, *p* = 0.012).

Attributes of final models: For colorectal cancer, the attributes of the final model were parish, year of presentation, vital status, sex, chemotherapy treatment, tumour dimension status, distant metastases, stated tumour extent, evidence of noncommunicable diseases other than cancer, number of care-related payments, and employment status at presentation (Table 3A), while for prostate cancer, these variables were age group, Gleason score, distant metastases, regional lymph node status, androgen deprivation therapy in the period, evidence of noncommunicable diseases other than cancer, number of care-related payments, and employment status at presentation (Table 3B).

## 4. Discussion

This is the first study in Antigua and Barbuda that examined whether disease stage is a predictor of time to drug therapy initiation (i.e., systemic therapy for colorectal cancer and androgen deprivation therapy for prostate cancer) for two common cancers affecting the local population. In this study, 36% of persons with colorectal cancer had drug therapy initiation within 90 days of being diagnosed, and 17% had therapy initiated after 90 days of being diagnosed. Regarding prostate cancer, 24% of cases received therapy after 90 days of being diagnosed/presentation, while 12% had therapy within 90 days of diagnosis. Where both cancers were concerned, disease stage on its own was not a positive predictor of time to drug therapy initiation. However, the final models generated for both cancers instead suggested that disease stage can only act in complement with other sample attributes, including distant metastases, evidence of noncommunicable diseases other than cancer, and the number of care-related payments, to produce a stable and significant model, with *p* < 0.05. Additionally, except for the other exposure variables which were significant in their respective models, disease stage was not found to be significant in either model.

The findings regarding the colorectal cancer model, though not suggestive of any ideal time to initiate systemic therapy, may yet be consistent with observations made in other studies which have pointed out that substantial delays in treatment for persons with some form of advanced disease could exacerbate disease progression, possibly leading to tumour recurrence or metastasis, as well as therapeutic failure [31]. While delays in initiating systemic therapy for distinct categories of colorectal cancer patients generally have implications for survival outcomes, our findings suggest that time to drug therapy initiation could easily be a reflection of other patient characteristics in addition to disease stage [31]. A study by Gao and colleagues pointed out that patient attributes, such as older age, cardiac issues, ostomy infection, acute renal failure, stroke, peritonitis, fistula of the gastrointestinal tract, shock, and septicaemia (whilst contributing to poorer patient outcomes, particularly if the patient was already compromised because of the disease process) could easily be affected by the toxicity of systemic therapy and thus would affect the time to initiation of drug therapy [31]. Future studies may want to consider the impact of specific issues, such as the type of surgery and/or postoperative complications, as well as their effect on the time to systemic drug therapy initiation and also on overall survival outcome [31].

Where prostate cancer is concerned, though the study findings do not provide any defined or ideal timespan for the initiation of androgen deprivation therapy, similar to the comments offered by Fossati et al. 2016, they nonetheless have much importance for what occurs in clinical practice locally [32]. To this end, the findings hint that any determination of disease stage and its relationship to the timeliness of initiation of androgen deprivation therapy must be consistent with the specific circumstances surrounding each case, including age, evidence of metastases, Gleason score, PSA level, receipt of radiation therapy, level of disease progression, and prognoses, among others [32,33]. Given that androgen deprivation therapy is part of the standard of care in prostate cancer management in Antigua and Barbuda, and that its use could significantly affect a patient’s quality of life and overall survival [34,35], deciding on when is the best time to introduce therapy—though hinging on specific attributes—could also easily be dependent on the healthcare facility and other patient challenges not disclosed by our study.

The odds ratios (ORs) derived from our final models point to the absence of statistical significance between disease stage and time to drug therapy initiation. This issue probably stemmed from our choice of sample sizes and the within-sample data variability [36]. Notwithstanding this and being cognizant of its possible effect on our hypothesis, however, the study’s findings still hold considerable clinical relevance when viewed in the context of the country’s socio-cultural experiences, existing challenges within the healthcare system, the cancer treatment models currently in use, and the ongoing epidemiological and economic burdens of the cancers studied [17,37,38]. This highlights the need for timely decisions regarding both the choice and appropriateness of drug treatment regimens linked to the management of colorectal and prostate cancers. For cancer care to be optimized, conscientious efforts, such as developing standardized colorectal and prostate cancer care algorithms, as well as offering training opportunities for clinicians and other healthcare personnel in the best evidence-based cancer care models, would be beneficial [17].

Currently, where both colorectal and prostate cancers are concerned, and despite recent advances in care, there is still no consensus regarding the optimal time to initiate drug therapy. The use of an expanded study period and data from an established local population-based or hospital-based registry could help any future evaluation of the relationship between disease stage and time to drug therapy initiation. This should be performed by considering the decision-making process regarding patient care, the involvement of a multidisciplinary team approach to care, and other nuances of the local health services that highlight quality of care issues [39].

Given the importance of drug therapy (systemic therapy for colorectal cancer and androgen deprivation therapy for prostate cancer) in improving patient outcomes for distinct categories of patients, including slowing disease progression and improving overall survival, our findings could, therefore, be considered a benchmark on whether specific attributes are evident when deciding to introduce drug therapy in our setting [40].

This study has several strengths. It has demonstrated the benefits of using retrospectively collected patient data to understand an aspect of the approach to initiating drug therapy for colorectal and prostate cancer patients in Antigua and Barbuda [41]. Data were also obtained from key establishments that account for a substantial proportion of documented evidence of cancer cases in Antigua and Barbuda [17]. Another strength is that this study provides baseline data and parameter estimates that may serve both to fill the gap created by the lack of a cancer registry and suggest the need for establishing one [17]. Additionally, the approach used by our study is instructive in offering a reasonable methodology that could be adapted for use in assessing the relationship between time to drug therapy initiation and disease stage from a prospective perspective in the foreseeable future. Prospective studies involving copious data derived over a lengthier timeframe and involving other intervals of time could be considered in the future. Another strength of this study was the availability of information on a broad range of variables. This encouraged a comprehensive and informative analysis that would not have been feasible had the study characteristics been reduced [42].

This study has some limitations that may have impacted its findings. Its retrospective study design suggests that it could easily have been affected by recall, recording, and social desirability biases [43], primarily if the information found on patient records, including symptom experience and the date of the first symptom, were not accurately described, interpreted, defined, or recorded at the time of a patient’s first consultation [43]. Furthermore, its retrospective design and reliance on specific data sources limited the availability of certain demographic, clinical, treatment, and socioeconomic information, leading to the inadvertent exclusion of several potential and important study variables. This issue could also have caused our models to be less predictive and/or contributed to the variations seen with respect to the crude and adjusted estimates derived for time to drug therapy initiation, especially when assessing our colorectal cancer models [44]. A larger-scaled prospective study that takes into account other important variables in a model evaluating the relationship between disease stage and time to drug therapy initiation in Antigua and Barbuda could be considered going forward. Moreover, the researchers used polytomous (multinomial) logistic regression in their analysis of the time to drug therapy initiation. This meant that the results could have been affected by the assumption that the categories of the time to drug therapy initiation were independent of each other, as well as the smallness of sample sizes for colorectal cancer cases in particular [45]. This invariably could account for the observed large confidence intervals for some of the categories of covariates included in the final colorectal cancer model. Notwithstanding this, however, we felt that our use of this mode of logistic regression (i) ensured that we maximized the use of the available data by considering all of the possible categories of time to drug therapy initiation and (ii) that we enabled a simplified interpretation of our results while also ensuring that our listed covariates were included as possible confounders in our modelling [24,46].

## 5. Conclusions

In conclusion, this study demonstrates that where colorectal and prostate cancers in Antigua and Barbuda are concerned, disease stage was not found to be a statistically significant predictor of time to drug therapy initiation in either cancer model. However, it may hold predictive value when assessed alongside other essential patient characteristics. These findings can inform efforts to optimize cancer care protocols in Antigua and Barbuda.

## Figures and Tables

**Table 1 healthcare-13-00915-t001:** Characteristics of colorectal and prostate cancer cases in Antigua and Barbuda (2017–2021), combined and disaggregated by time to drug therapy initiation given in months.

Characteristics	Colorectal Cancer (N = 70) %	Time to Systemic Drug Therapy Initiation			Prostate Cancer (N = 109) %	Time to Androgen Drug Therapy Initiation		
		No Therapy Administered N = 33	Early Time (≤3 Months) N = 25	Late Time (>3 Months) N = 12		Not Stated/Unknown N = 70	Early Time (≤3 Months) N = 13	Late Time (>3 Months) N = 26
Age at Presentation								
Mean age (±SD)	66.30 (11.72)	67.15 (12.40)	65.60 (9.91)	65.42 (13.98)	66.47 (7.77)	66.70 (7.55)	65.00 (8.29)	66.58 (8.34)
Mean age 95%CI	63.51–69.09	62.75–71.55	61.51–69.69	56.53–74.30	65.00–67.94	64.90–68.50	60.00–70.00	63.21–69.95
Median age (IQR)	68.50 (20.00)	70.00 (18.00)	67.00 (15.00)	69.00 (23.00)	67.00 (11.00)	67.00 (11.00)	62.00 (14.00)	68.00 (10.00)
Range	42.00–87.00	42.00–87.00	43.00–78.00	42.00–87.00	46.00–90.00	51.00–90.00	55.00–80.00	46.00–80.00
Age-group (Years)								
<50	4 (5.71)	2 (6.06)	1 (4.00)	1 (8.33)	n/a	n/a	n/a	n/a
50–60	18 (25.71)	9 (27.27)	6 (24.00)	3 (25.00)	n/a	n/a	n/a	n/a
60–70	16 (22.86)	5 (15.15)	8 (32.00)	3 (25.00)	n/a	n/a	n/a	n/a
>70	32 (45.71)	17 (51.52)	10 (40.00)	5 (41.67)	n/a	n/a	n/a	n/a
Age group (Years)								
<60	n/a	n/a	n/a	n/a	18 (16.51)	11 (15.71)	4 (30.77)	3 (11.54)
60–70	n/a	n/a	n/a	n/a	53 (48.62)	35 (50.00)	5 (38.46)	13 (50.00)
>70	n/a	n/a	n/a	n/a	38 (34.86)	24 (34.29)	4 (30.77)	10 (38.46)
Parish (area of residence)								
All other parishes (rural area)	33 (47.14)	12 (36.36)	14 (56.00)	7 (58.33)	41 (37.61)	23 (32.86)	4 (30.77)	14 (53.85)
St. John’s (urban area)	37 (52.86)	21 (63.64)	11 (44.00)	5 (41.67)	68 (62.39)	47 (67.14)	9 (69.23)	12 (46.15)
Year of Presentation								
2017	10 (14.29)	6 (18.18)	3 (12.00)	1 (8.33)	11 (10.09)	8 (11.43)	1 (7.69)	2 (7.69)
2018	13 (18.57)	7 (21.21)	4 (16.00)	2 (16.67)	11 (10.09)	6 (8.57)	2 (15.38)	3 (11.54)
2019	7 (10.00)	4 (12.12)	2 (8.00)	1 (8.33)	27 (24.77)	18 (25.71)	4 (30.77)	5 (19.23)
2020	26 (36.14)	10 (30.30)	10 (40.00)	6 (50.00)	40 (36.70)	22 (31.43)	4 (30.77)	14 (53.85)
2021	14 (20.00)	6 (18.18)	6 (24.00)	2 (16.67)	20 (18.35)	16 (22.86)	2 (15.38)	2 (7.69)
COVID-19 Effect								
Pre-COVID-19	30 (42.86)	17 (51.52)	9 (36.00)	4 (33.33)	49 (44.95)	32 (45.71)	7 (53.85)	10 (38.46)
During COVID-19	40 (57.14)	16 (48.48)	16 (64.00)	8 (66.67)	60 (55.05)	38 (54.29)	6 (46.15)	16 (61.54)
Vital Status								
Died	18 (25.71)	7 (21.21)	7 (28.00)	4 (33.33)	13 (11.93)	9 (12.86)	2 (15.38)	2 (7.69)
Alive	52 (74.29)	26 (78.79)	18 (72.00)	8 (66.67)	96 (88.07)	61 (87.14)	11 (84.62)	24 (92.31)
**Sex**								
Female	37 (52.86)	18 (54.55)	12 (48.00)	7 (58.33)	n/a	n/a	n/a	n/a
Male	33 (47.14)	15 (45.45)	13 (52.00)	5 (41.67)	n/a	n/a	n/a	n/a
Disease Stage								
Early-Stage 1/2	29 (41.43)	15 (45.45)	10 (40.00)	4 (33.33)	59 (54.13)	36 (51.43)	7 (53.85)	16 (61.54)
Late-Stage 3/4	41 (58.57)	18 (54.55)	15 (60.00)	8 (66.67)	50 (45.87)	34 (48.57)	6 (46.15)	10 (38.46)
Histological Grade								
Grade1	8 (11.43)	6 (18.18)	2 (8.00)	0	n/a	n/a	n/a	n/a
Grade2	58 (82.86)	27 (81.82)	21 (84.00)	10 (83.33)	n/a	n/a	n/a	n/a
Grade3	4 (5.71)	0	2 (8.00)	2 (16.67)	n/a	n/a	n/a	n/a
Tumour Site								
Undetermined	6 (8.57)	3 (9.09)	3 (12.00)	0	n/a	n/a	n/a	n/a
Determined	64 (91.43)	30 (90.91)	22 (88.00)	12 (100.00)	n/a	n/a	n/a	n/a
Chemotherapy Treatment								
No	35 (50.00)	26 (78.79)	4 (16.00)	5 (41.67)	n/a	n/a	n/a	n/a
Yes	35 (50.00)	7 (21.21)	21 (84.00)	7 (58.33)	n/a	n/a	n/a	n/a
Tumour Dimension Status								
Undetermined	24 (34.29)	11 (33.33)	9 (36.00)	4 (33.33)	n/a	n/a	n/a	n/a
≤5 cm	24 (34.29)	12 (36.36)	6 (24.00)	6 (50.00)		n/a	n/a	n/a
>5 cm	22 (31.43)	10 (30.30)	10 (40.00)	2 (16.67)	n/a	n/a	n/a	n/a
Gleason Score								
Not Stated	n/a	n/a	n/a	n/a	20 (18.35)	12 (17.14)	2 (15.38)	6 (23.08)
6 to 7	n/a	n/a	n/a	n/a	61 (55.96)	40 (57.14)	7 (53.85)	14 (53.85)
8 to 10	n/a	n/a	n/a	n/a	28 (25.69)	18 (25.71)	4 (30.77)	6 (23.08)
Distant Metastases								
Undetermined (Mx/Not Stated)	29 (41.43)	17 (51.52)	9 (36.00)	3 (25.00)	85 (77.98)	64 (91.43)	6 (46.15)	15 (57.69)
Determined (M0/M1)	41 (58.57)	16 (48.48)	16 (64.00)	9 (75.00)	24 (22.02)	6 (8.57)	7 (53.85)	11 (42.31)
Regional Lymph Node Status								
Undetermined (Nx/Not Stated)	7 (10.00)	5 (15.15)	1 (4.00)	1 (8.33)	80 (73.39)	61 (87.14)	6 (46.15)	13 (50.00)
Determined (N0/N2)	63 (90.00)	28 (84.85)	24 (96.00)	11 (91.67)	29 (26.61)	9 (12.86)	7 (53.85)	13 (50.00)
Primary Tumour								
Undetermined	7 (10.00)	5 (15.15)	1 (4.00)	1 (8.33)	78 (71.56)	59 (84.29)	7 (53.85)	12 (46.15)
Determined (T1/T4)	63 (90.00)	28 (84.85)	24 (96.00)	11 (91.67)	31 (28.44)	11 (15.71)	6 (46.15)	14 (53.85)
Androgen Deprivation Therapy in Period								
No	n/a	n/a	n/a	n/a	39 (35.78)	27 (38.57)	2 (15.38)	10 (38.46)
Yes	n/a	n/a	n/a	n/a	70 (64.22)	43 (61.43)	11 (84.62)	16 (61.54)
Radiation Therapy Status								
Incomplete/Unknown	n/a	n/a	n/a	n/a	53 (48.62)	35 (50.00)	4 (30.77)	14 (53.85)
Completed/known	n/a	n/a	n/a	n/a	56 (51.38)	35 (50.00)	9 (69.23)	12 (46.15)
Stated Tumour Extent								
No	19 (27.14)	13 (39.39)	3 (12.00)	3 (25.00)	n/a	n/a	n/a	n/a
Yes	51 (72.86)	20 (60.61)	22 (88.00)	9 (75.00)	n/a	n/a	n/a	n/a
Evidence of Noncommunicable Disease other than Cancer								
No	36 (51.43)	19 (57.58)	11 (44.00)	6 (50.00)	67 (61.47)	38 (54.29)	11 (84.62)	18 (69.23)
Yes	34 (48.57)	14 (42.42)	14 (56.00)	6 (50.00)	42 (38.53)	32 (45.71)	2 (15.38)	8 (30.77)
No. of Care-Related Payments in Period								
≤10	34 (48.57)	22 (66.67)	8 (32.00)	4 (33.33)	55 (50.46)	31 (44.29)	9 (69.23)	15 (57.69)
10–20	20 (28.57)	10 (30.30)	8 (32.00)	2 (16.67)	29 (26.61)	21 (30.00)	3 (23.08)	5 (19.23)
>20	16 (22.86)	1 (3.03)	9 (36.00)	6 (50.00)	25 (22.94)	18 (25.71)	1 (7.69)	6 (23.08)
Employment Status at Time of Presentation								
Not employed	31 (44.29)	16 (48.48)	7 (28.00)	8 (66.67)	29 (26.61)	21 (30.00)	2 (15.38)	6 (23.08)
Employed	39 (55.71)	17 (51.52)	18 (72.00)	4 (33.33)	80 (73.39)	49 (70.00)	11 (84.62)	20 (76.92)
Time to Drug Therapy Initiation (months)								
No therapy administered	33 (47.14)	33 (100.00)	0	0	n/a	n/a	n/a	n/a
Early Time (≤3 months)	25 (35.71)	0	25 (100.00)	0	n/a	n/a	n/a	n/a
Late Time (>3 months)	12 (17.14)	0	0	12 (100.00	n/a	n/a	n/a	n/a
Time Interval to Androgen Drug Therapy Initiation (months)								
Not stated/unknown	n/a	n/a	n/a	n/a	70 (64.22)	70 (100.00)	0	0
Early Time (≤3 months)	n/a	n/a	n/a	n/a	13 (11.93)	0	13 (100.00)	0
Late Time (>3 months)	n/a	n/a	n/a	n/a	26 (23.85)	0	0	26 (100.00)

n/a: not applicable.

**Table 2 healthcare-13-00915-t002:** (**A**) The results of the bivariate multinomial logistic regression analysis for colorectal cancer. (**B**) The results of the bivariate multinomial logistic regression analysis for prostate cancer.

(A)			
Characteristics	Bivariable Analysis		
	Odds Ratio (95% CI) Early Time (≤3 months)	Odds Ratio (95% CI) Late Time (>3 months)	Model *p*-Value
**Age at Presentation**			
Mean age (±SD)	0.99 (0.95–1.03)	0.99 (0.93–1.04)	0.85
**Age Group (Years)**			
<50	Ref	Ref	0.86
50–60	1.33 (1.00–18.19)	0.67 (0.04–10.25)	
60–70	3.20 (0.23–45.19)	1.20 (0.07–19.63)	
>70	1.18 (0.09–14.69)	0.59 (0.04–7.91)	
**Parish (Area of Residence)**			
All other parishes (rural area)	Ref	Ref	0.23
St. John’s (urban area)	0.45 (0.15–1.30)	0.41 (0.11–1.57)	
**Year of Presentation**			
2017	Ref	Ref	0.96
2018	1.14 (0.18–7.28)	1.71 (0.12–23.94)	
2019	1.0 (0.11–8.95)	1.50 (0.07–31.57)	
2020	2.00 (0.39–10.31)	3.60 (0.34–37.62)	
2021	2.00 (0.34–11.97)	2.00 (0.14–28.42)	
**COVID-19 Effect**			
Pre-COVID-19	Ref	Ref	0.38
During COVID-19	1.89 (0.65–5.48)	2.13 (0.53–8.45)	
**Vital Status**			
Died	Ref	Ref	0.68
Alive	0.69 (0.21–2.32)	0.54 (0.12–2.32)	
**Sex**			
Female	Ref	Ref	0.81
Male	1.30 (0.46–3.68)	0.86 (0.23–3.26)	
**Disease Stage**			
Early-Stage 1/2	Ref	Ref	0.75
Late-Stage 3/4	1.25 (0.44–3.58)	1.67 (0.42–6.64)	
**Chemotherapy Treatment** **			
No	Ref	Ref	**<0.001**
Yes	**19.50 (5.02–75.70)**	**5.20 (1.26–21.50)**	
**Tumour Dimension Status**			
Undetermined	Ref	Ref	0.52
≤5 cm	0.61 (0.16–2.28)	1.38 (0.30–6.20)	
>5 cm	1.22 (0.35–4.24)	0.55 (0.08–3.68)	
**Distant Metastases**			
Undetermined (Mx/Not Stated)	Ref	Ref	0.21
Determined (M0/M1)	1.89 (0.65–5.48)	3.19 (0.73–13.92)	
**Regional Lymph Node Status**			
Undetermined (Nx/Not Stated)	Ref	Ref	0.34
Determined (N0/N2)	4.29 (0.47–39.27)	1.96 (0.21–18.78)	
**Primary Tumour**			
Undetermined	Ref	Ref	0.34
Determined (T1/T4)	4.29 (0.47–39.27)	1.96 (0.21–18.78)	
**Stated Tumour Extent ***			
No	Ref	Ref	**0.06**
Yes	**4.77 (1.18–19.21)**	1.95 (0.44–8.58)	
**Evidence of Noncommunicable Disease other than Cancer**			
No	Ref	Ref	
Yes	1.73 (0.61–4.93)	1.36 (0.36–5.11)	
**No. of Care-Related Payments in Period** **			
≤10	Ref	Ref	**<0.001**
10–20	2.20 (0.64–7.55)	1.10 (0.17–7.03)	
>20	**24.75 (2.69–227.61)**	**33.00 (3.09–353.00)**	
**Employment Status at Time of Presentation**			
Not employed	Ref	Ref	**0.06**
Employed	2.42 (0.80–7.33)	0.47 (0.12–1.87)	
**Time to Drug Therapy Initiation (months)**			
No therapy administered	n/a	n/a	
Early Time (≤3 months)	n/a	n/a	
Late Time (>3 months)	n/a	n/a	
**(B)**			
**Characteristics**	**Bivariable Analysis**		
	**Odds Ratio (95% CI)** **Early Time (≤ 3 Months)**	**Odds Ratio (95% CI)** **Late Time (> 3 Months)**	**Model *p*-Value**
**Age at Presentation**			
Mean age (±SD)	0.97 (0.90–1.04)	1.00 (0.94–1.06)	0.76
**Age Group (Years)**			
<60	Ref	Ref	
60–70	0.39 (0.09–1.72)	1.36 (0.33–5.67)	
>70	0.46 (0.10–2.18)	1.53 (0.35–6.67)	
**Parish (Area of Residence)**			
All other parishes (rural area)	Ref	Ref	0.15
St. John’s (urban area)	1.10 (0.31–3.96)	0.42 (0.17–1.05)	
**Year of Presentation**			
2017	Ref	Ref	0.56
2018	2.67 (0.19–36.76)	2.00 (0.25–16.00)	
2019	1.78 (0.17–18.53)	1.11 (0.18–7.00)	
2020	1.45 (0.14–15.04)	2.55 (0.47–13.77)	
2021	1.00 (0.08–12.76)	0.50 (0.06–4.23)	
**COVID-19 Effect**			
Pre-COVID-19	Ref	Ref	
During COVID-19	0.72 (0.22–2.37)	1.35 (0.54–3.38)	
**Vital Status**			
Died	Ref	Ref	0.71
Alive	0.81 (0.15–4.27)	1.77 (0.36–8.80)	
**Disease Stage**			
Early-Stage 1/2	Ref	Ref	0.67
Late-Stage 3/4	0.91 (0.28–2.97)	0.66 (0.26–1.66)	
**Gleason Score**			
Not Stated	Ref	Ref	0.96
6 to 7	1.05 (0.19–5.74)	0.70 (0.22–2.22)	
8 to 10	1.33 (0.21–8.46)	0.67 (0.17–2.56)	
**Distant Metastases** **			
Undetermined (Mx/Not Stated)	Ref	Ref	**<0.001**
Determined (M0/M1)	**12.44 (3.15–49.19)**	**7.82 (2.50–24.52)**	
**Regional Lymph Node Status** **			
Undetermined (Nx/Not Stated)	Ref	Ref	**<0.001**
Determined (N0/N2)	**7.91 (2.16–28.89)**	**6.78 (2.40–19.17)**	
**Primary Tumour** **			
Undetermined	Ref	Ref	**<0.001**
Determined (T1/T4)	**4.60 (1.30–16.31)**	**6.26 (2.29–17.09)**	
**Androgen Deprivation Therapy in Period**			
No	Ref	Ref	0.22
Yes	3.45 (0.71–16.79)	1.00 (0.40–2.53)	
**Radiation Therapy status**			
Incomplete/Unknown	Ref	Ref	0.36
Completed/known	2.25 (0.63–8.00)	0.86 (0.35–2.11)	
**Evidence of Noncommunicable Disease other than Cancer**			
No	Ref	Ref	
Yes	0.22 (0.04–1.05)	0.53 (0.20–1.37)	
**No. of Care-Related Payments in Period**			
≤10	Ref	Ref	0.33
10–20	0.49 (0.12–2.03)	0.49 (0.16–1.56)	
>20	0.19 (0.02–1.64)	0.69 (0.23–2.09)	
**Employment Status at Time of Presentation**			
Not employed	Ref	Ref	0.47
Employed	2.36 (0.48–11.57)	1.43 (0.50–4.07)	
**Time to Androgen Drug Therapy Initiation (months)**			
Not stated/unknown	n/a	n/a	n/a
Early Time (≤3 months)	n/a	n/a	n/a
Late Time (>3 months)	n/a	n/a	n/a

* The variable shows evidence of confounding the relationship between time to drug therapy initiation and disease stage. ** The variable shows significance in the bivariate model (*p* < 0.05).

**Table 3 healthcare-13-00915-t003:** (**A**) The results of the polytomous multivariable logistic regression analysis on disease stage, time to drug therapy initiation, and key covariates of the final model for colorectal cancer. (**B**) The results of the polytomous multivariable logistic regression analysis on disease stage, time to drug therapy initiation, and key covariates of the final model for prostate cancer.

(A)		
Characteristics	Multivariable Analysis	
	Adjusted Odds Ratio (95% CI) Early Time ≤3 months	Adjusted Odds Ratio (95% CI) Late Time >3 months
Colorectal cancer—Overall *p*-Value < 0.001		
Disease Stage		
Early-Stage 1/2	Ref	Ref
Late-Stage 3/4	1.25 (0.09–17.92)	41.58 (0.78–2219.28)
Parish (Area of Residence)		
All other parishes (rural area)	Ref	Ref
St. John’s (urban area)	0.42 (0.03–5.96)	0.28 (0.01–5.55)
Year of Presentation*		
2017	Ref	Ref
2018	63.87 (0.18–22,228.55)	**2816.41 (4.07–1,948,183)**
2019	811.29 (0.34–1,963,492)	1065.54 (0.17–6,552,967)
2020	22.75 (0.07–7106.73)	**2478.28 (3.43–1,790,327)**
2021	0.24 (0.001–77.71)	396.08 (0.41–378,806.10)
Vital Status		
Died	Ref	Ref
Alive	0.19 (0.01–4.40)	0.73 (0.03–17.94)
Sex		
Female	Ref	Ref
Male	3.61 (0.31–42.73)	2.15 (0.12–39.83)
Chemotherapy Treatment *		
No	Ref	Ref
Yes	**2236.14 (14.54–344,007.80)**	**117.93 (1.09–12,744.34)**
Tumour Dimension Status		
Undetermined	Ref	Ref
≤5 cm	0.03 (0.001–1.05)	0.08 (0.002–3.62)
>5 cm	0.68 (0.02–19.95)	0.11 (0.003–4.19)
Distant Metastases		
Undetermined (Mx/Not Stated)	Ref	Ref
Determined (M0/M1)	0.24 (0.02–3.12)	0.26 (0.001–7.09)
Stated Tumour Extent		
No	Ref	Ref
Yes	**3701.19 (11.33–1,208,649)**	4.55 (0.06–361.86)
Evidence of Noncommunicable Disease other than Cancer		
No	Ref	Ref
Yes	19.77 (0.51–764.51)	2.00 (0.07–57.12)
No. of Care-Related Payments in Period *		
≤10	Ref	Ref
10–20	0.69 (0.05–8.94)	0.18 (0.01–4.27)
>20	**274.76 (2.08–36,283.61)**	**940.31 (5.14–172,179.80)**
Employment Status at Time of Presentation *		
Not employed	Ref	Ref
Employed	**38.96 (1.45–1048.92)**	0.21 (0.01–6.88)
**(B)**		
**Characteristics**	**Multivariable Analysis**	
	**Adjusted Odds Ratio (95% CI)** **Early Time ≤3 months**	**Adjusted Odds Ratio (95% CI) Late Time >3 months**
Prostate Cancer—Overall *p*-Value = 0.019		
Disease Stage		
Early-Stage 1/2	Ref	Ref
Late-Stage 3/4	0.19 (0.03–1.13)	0.41 (0.11–1.44)
Age Group (Years)		
<60	Ref	Ref
60–70	0.35 (0.05–2.27)	1.35 (0.26–7.13)
>70	1.66 (0.17–16.38)	2.60 (0.41–16.60)
Gleason Score		
Not Stated	Ref	Ref
6 to 7	1.32 (0.15–11.81)	0.60 (0.15–2.47)
8 to 10	1.92 (0.18–20.75)	0.87 (0.18–4.23)
Distant Metastases *		
Undetermined (Mx/Not Stated)	Ref	Ref
Determined (M0/M1)	**56.94 (1.15–2827.86)**	4.28 (0.55–33.11)
Regional Lymph Node Status		
Undetermined (Nx/Not Stated)	Ref	Ref
Determined (N0/N2)	0.45 (0.01–15.58)	2.74 (0.44–17.11)
Androgen Deprivation Therapy in Period		
No	Ref	Ref
Yes	4.93 (0.66–37.13)	1.19 (0.35–4.03)
Evidence of Noncommunicable Disease other than Cancer		
No	Ref	Ref
Yes	0.24 (0.03–1.89)	0.55 (0.17–1.80)
No. of Care-Related Payments in Period		
≤10	Ref	Ref
10–20	0.37 (0.06–2.34)	0.32 (0.07–1.39)
>20	0.29 (0.02–3.51)	0.86 (0.23–3.28)
Employment Status at Time of Presentation		
Not employed	Ref	Ref
Employed	4.87 (0.39–61.28)	1.39 (0.30–6.34)

* The variables or variable categories with odds ratios (ORs) in bold were significant in the final model (*p* < 0.05). Colorectal cancer: overall *p*-value of model (*p* < 0.001). Prostate cancer: overall *p*-value of model (*p* = 0.019).

## Data Availability

The data generated for this study are from the Sir Lester Bird Medical Centre, Medical Benefits Scheme, The Eastern Caribbean Cancer Centre and Health Information Division, Ministry of Health, Antigua and Barbuda and cannot be made publicly available. All interested readers can access the data through the Antigua and Barbuda Institutional Review Board, Ministry of Health, the Institutional Review Board of the Sir Lester Bird Medical Centre, and the University of KwaZulu-Natal Biomedical Research Ethics Committee (BREC) from the following contacts: Chairperson, Antigua and Barbuda Institutional Review Board, Ministry of Health, Antigua and Barbuda, Tel: (1-268-464-5685), (1-268-462-5685); Medical Director, Sir Lester Bird Medical Centre, Antigua and Barbuda, Tel: (1-268-484-2700), (1-268-484-2788), Fax: (1-268-484-2702); Chair, Biomedical Research Ethics Committee, University of KwaZulu-Natal Research Ethics Office, Westville Campus, Govan Mbeki Building, Private Bag X54001, Durban, 4000, KwaZulu-Natal, South Africa, Tel.: +27 31 260 4769, Fax: +27 31 260 4609, e-mail: BREC@ukzn.ac.za.

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
