# Peer review of "The Timeliness of Drug Therapy in Colorectal and Prostate Cancer in Antigua and Barbuda: The Role of Disease Stage"

_healthcare, 2025, doi:10.3390/healthcare13080915_

Round 1
Reviewer 1 Report
Comments and Suggestions for Authors
Areas for Improvement:
1. Clarity and Language:
The review contains awkward phrasing and minor grammatical issues. Examples:
“Analyses was performed…” → should be “Analyses were performed…”
“Yet disease stage was not found…” → stylistically better as “However, disease stage was not found…”
“In a model depicting other important patient characteristics.” → This is vague and needs clarification or rephrasing.
2. Structure and Flow:
The transitions between sections are abrupt. For instance, the conclusion could be more logically connected to the results and implications.
The sentence “Though final models were significant… yet disease stage was not…” is confusing. Consider simplifying or splitting this for clarity.
3. Methodological Details:
The study mentions four data collection sites but provides no detail on sample size, data quality, or how disease stage and time-interval were defined.
The wide confidence interval for colorectal cancer (95% CI 0.78–2219.28) raises questions about sample size or data variability, which should be briefly acknowledged.
4. Interpretation of Results:
The conclusion should be more precise. The abstract states that disease stage is not a predictor “except in a model depicting other characteristics,” but this model is not described. Consider summarizing these other variables briefly or stating the model limitations more clearly.
5. Impact Statement:
Suggested Revision:
Improve grammar and readability
Clarify key findings and their statistical significance
Briefly define the patient population and data
Make the conclusion more cohesive with results and methodology
Clarity and Language:
The review contains awkward phrasing and minor grammatical issues. Examples:
“Analyses was performed…” → should be “Analyses were performed…”
“Yet disease stage was not found…” → stylistically better as “However, disease stage was not found…”
“In a model depicting other important patient characteristics.” → This is vague and needs clarification or rephrasing.
Author Response
|
Response to Reviewer 1 Comments
|
||
|
1. Summary |
|
|
|
Thank you very much for taking the time to review this manuscript. We do express our appreciations to you for your comments and suggestions offered. It is our hope that the revised manuscript has addressed your concerns. We do look forward to hearing from you on this. Please find the detailed responses below and the corresponding revisions/corrections highlighted in track changes in the re-submitted files.
|
||
|
2. Point-by-point response to Comments and Suggestions for Authors
|
||
|
Comment (1) Clarity and Language: The review contains awkward phrasing and minor grammatical issues. Examples: Response (1) The authors have taken a keen note of the reviewer’s comments. To this end we made the following edits to the areas highlighted by the reviewer: See lines: Line 19 Analyses were performed..
however, disease stage was not... Regarding both cancers, our findings demonstrated that disease stage alone is not a significant predictor of the time-interval to the initiation of drug therapy, unless analyzed alongside other essential patient characteristics in each respective model. Additionally, we did subject the manuscript to English language editing to correct and/or improve on phrasing and grammar issues. This was done by the corresponding author.
Comment (2) Structure and Flow: The transitions between sections are abrupt. For instance, the conclusion could be more logically connected to the results and implications.
Response (2) The authors do wish to thank the reviewer for pointing out these issues. To improve readability and remove any ambiguity, we have edited the appropriate section in the abstract. This now reads See lines 20-22 Analyses showed that the final models for both cancers were significant (P<0.05), however, disease stage was not found to be a predictor of time-interval to drug therapy initiation in either model. ORs observed were colorectal cancer 41.58 (95% CI 0.78-2219.28) and prostate cancer 0.41 (95% CI 0.11-1.44). Additionally, we did extensive editing to the manuscript to correct and/or improve grammar and phrasing issues.
Comment (3) The study mentions four data collection sites but provides no detail on sample size, data quality, or how disease stage and time-interval were defined.
Response (3) The authors have taken a keen note of the reviewer’s comments and wish to point out the following: Regarding sample size: On the matter of sample size the authors wish to share that this was done on the basis of convenience as the study attempted to consider all cases of both colorectal and prostate cancers that were ever diagnosed between 1 January 2017 to 31 December 2021. Notwithstanding this, and so as not to lose the importance of the reviewer’s comment or suggestion, the authors have since inserted a subhead titled sample size which addresses this matter by way of a short sentence. See lines 87-90 Consistent with the study’s use of data reported in a previous publication [15], the study’s objective and because of convenience, we sought to utilize data for all cases diagnosed with colorectal (n=79) and prostate cancer (n=109), respectively. Regarding data quality: Concerning this point raised by the reviewer, the authors wish to share that on the basis that this study utilized data and results previously reported in the published article “Incidence, trends and patterns of female breast, cervical, colorectal and prostate cancers in Antigua and Barbuda, 2017–2021: a retrospective study” we sought to avoid repetitiveness by acknowledging under the subhead ‘Study design, setting and population’ that we used said data and some results. To this end we did not include a section on variables and data preparation specifically. We did however mention under subhead ‘Data collection and management’ the following which speaks broadly to data quality issues (relevance, completeness, reliability, accuracy): See lines 93-94 As mentioned in an earlier study [15], data used in this study were extracted from patient records of persons diagnosed with colorectal or prostate cancer in accordance with the International Classification of Diseases Tenth Edition (ICD-10) (C18, C19, C20-colon/rectum) and (C61-prostate) [16], and archived with either the Oncology, Pathology and Urology departments of the Sir Lester Bird Medical Centre (SLBMC), and The Cancer Centre Eastern Caribbean, Antigua and Barbuda (TCCEC) [15]. We collected additional data on patients from the Medical Benefits Scheme (MBS) and data on patient deaths from the Health Information Division, Ministry of Health, Antigua and Barbuda [15]. Regarding how disease stage was defined: The authors wish to share that we did include in brief the definition of disease stage under the subhead titled “Outcome Ascertained”. Under this heading disease stage is given as See lines 115-116 The main exposure was disease stage at diagnosis, a dichotomous variable with groups ‘Early-stage disease (clinical stages 1 and 2)’ and ‘Late-stage disease (clinical stages 3 and 4)’. Regarding how time-interval was defined: The authors wish to share that we did include in brief the definition of time-interval under the subhead titled “Outcome Ascertained”. Under this heading time-interval is given as See lines 117-119 The main outcome was time-interval to the initiation of systemic drug therapy, a categorical variable divided into three (3) groups: (i) no drug therapy in the period, (ii) early time-interval (≤ 3 months), and (iii) late time-interval (> 3 months).
Comment (4) The wide confidence interval for colorectal cancer (95% CI 0.78–2219.28) raises questions about sample size or data variability, which should be briefly acknowledged. Response (4) The authors have taken a keen note of the reviewer’s comment and wish to share that we have since acknowledged this observation in relation to a possible limitation of using multinomial logistic regression. We have thus included a few sentences in the discussion section under study limitations that speak to this matter. See lines 441-449 Moreover, the researchers used polytomous (multinomial) logistic regression in their analysis of time-interval to drug therapy initiation, this meant that the results could have been affected by the assumption that categories of time-interval to drug therapy initiation are independent of each other as well as the smallness of sample sizes for colorectal cancer cases in particular [39]. This invariably could account for the observed large confidence intervals for some categories of covariates included in the final colorectal cancer model. Notwithstanding this however, we felt that our use of this mode of logistic regression ensured that we maximized use of the available data by considering all possible categories of time-interval to drug therapy, enabled a simplified interpretation of our results while also ensuring that we included our listed covariates as possible confounders in our modelling [40,41]. The authors do hope that we have sufficiently addressed the reviewer’s comments. |
||
|
Comment (5) . Interpretation of Results: The conclusion should be more precise. The abstract states that disease stage is not a predictor “except in a model depicting other characteristics,” but this model is not described. Consider summarizing these other variables briefly or stating the model limitations more clearly. Response (5 ) The authors have taken a keen note of the reviewer’s comments and to this end we have inserted under the subhead ‘multivariable analyses’ a summary statement that captures all the attributes or variables belonging to the requisite final models of colorectal and prostate cancers, respectively.
See lines 356-362 Attributes of final models: For colorectal cancer, the attributes of the final model were parish, year of presentation, vital status, sex, chemotherapy treatment, tumour dimensions status, distant metastases, stated tumour extent, evidence of noncommunicable disease other than cancer, number of cancer-related payments and employment status at presentation (Table 3A), while for prostate cancer these variables were age-group, Gleason score, distant metastases, regional lymph node status, androgen deprivation therapy in the period, evidence of noncommunicable disease other than cancer, number of care related payments, and employment status at presentation (Table 3B).
Additionally, and so that readers are provided sufficient clarity when interpreting our findings, we have inserted as part of the study limitations the following:
See lines 437-439 This issue could also have caused our models to be less predictive and/or contributed to the variations seen in respect of the crude and adjusted estimates derived for time-interval to drug therapy initiation, especially when assessing our colorectal cancer models [39].
|
||
|
Comment (6) Impact Statement: Suggested Revision:
Response (6)
Improve grammar and readability – This was done by way of extensive English language done by the corresponding author.
Clarify key findings and their statistical significance – This was done by summarizing our larger tables to present only crude and adjusted estimates; by including a final paragraph under subhead ‘multivariable analyses’ that addressed the variables in the requisite final models and by editing sections of the study limitations to include a few key points.
Briefly define the patient population and data -The authors wish to point out that this was previously stated under subhead “Study design, setting and population”
Additionally, we have included a subhead titled ‘sample size’ to further address this matter.
We have edited the ‘Conclusions’ section of the manuscript to read thus:
See lines 451-462
In conclusion, this study demonstrated that where colorectal and prostate cancers in Antigua and Barbuda are concerned, disease stage on its own is neither a definitive nor significant predictor of the time-interval to the initiation of drug therapy, except, if analyzed alongside other model attributes such as distant metastases, evidence of noncommunicable disease other than cancer and number of care related payments, among others. Making improvements to local data repositories and conducting future studies to assess things like (i) the relationship between time-interval to drug therapy initiation and disease stage through the use of prospectively collected data and a wider array of population attributes, (ii) the effect of type of surgery, postoperative complications and their effect on time-interval to the initiation of systemic drug therapy, and (iii) the effect of time-interval to drug therapy initiation on either overall survival or progression-free survival for patients diagnosed with colorectal or prostate cancers are recommended. Furthermore, and despite the mixed study results, our findings may act as a valuable resource for policymakers aiming to improve screening, diagnostic and treatment capabilities locally, while also seeking to establish a standardized care algorithm that caters to the timely initiation of drug treatment for specific categories of persons diagnosed with these cancers in the foreseeable future.
|
||
|
Kindly note that in addition to the edits done in respect of the comments and/or suggestions of the Reviewer, the authors have made some edits to further improve the article and so as to ensure that there is consistency across all areas of our study re: scope and/or purpose. This included edits to text and tables. |
||
|
|
||
|
Thank you |
||

Reviewer 2 Report
Comments and Suggestions for Authors
- The novelty must be highlighted in the abstract and at the end of introduction, especially relative to the article: “Rhudd AR. The current state of prostate cancer in Antigua & Barbuda-2021. Ecancermedicalscience. 2021 Aug 17;15:ed112. doi: 10.3332/ecancer.2021.ed112. PMID: 34567266; PMCID: PMC8426014.”.
- In the sentence: “In 35 2022 there were approximately 20 million new cancer cases and around 9.7 million cancer-36 related deaths”, please replace it with newer statistics for cancer. You strongly suggested updating the whole paragraph to 2025 statistics.
- The COVID-19 effect should be briefly included in the abstract since it was treated in the article.
- In Table 1, please indicate the unit of measurement (i.e., month, year....etc).
- Tables 2 A and B are too tedious for the reader and must be summarized into one more brief table, which also applies to Tables 3 A and B.
Author Response
|
Response to Reviewer 2 Comments
|
||
|
1. Summary |
|
|
|
Thank you very much for taking the time to review this manuscript. We do express our appreciations to you for your comments and suggestions offered. It is our hope that the revised manuscript has addressed your concerns. We do look forward to hearing from you on this. Please find the detailed responses below and the corresponding revisions/corrections highlighted in track changes in the re-submitted files.
|
||
|
2. Point-by-point response to Comments and Suggestions for Authors
|
||
|
Comment (1) The novelty must be highlighted in the abstract and at the end of introduction, especially relative to the article: “Rhudd AR. The current state of prostate cancer in Antigua & Barbuda-2021. Ecancermedicalscience. 2021 Aug 17;15:ed112. doi: 10.3332/ecancer.2021.ed112. PMID: 34567266; PMCID: PMC8426014.”.
Response (1) The authors have taken note of the reviewer’s comment and wish to share that we have since improved the final paragraph of the introduction section by inserting the following words:
See lines 71-73 On the basis that the systematic collection of data regarding colorectal and prostate cancers will help in providing both objective insights and measurable evidence of Antigua and Barbuda’s colorectal and prostate cancer management capabilities [16], Additionally, we have also cited the article “Rhudd 2021” PMID: 34567266; PMCID: PMC8426014.” to support this edit.
Further we have also edited the abstract to show that we have accounted for this point-highlighting novelty-
See lines 12-15 In Antigua and Barbuda, where resources are limited, there is a need for both insight and evidence on the timeliness of drug therapy initiation for colorectal and prostate cancers as a way of improving disease management capabilities and prognostic outcomes for diagnosed cases. Comment (2) In the sentence: “In 35 2022 there were approximately 20 million new cancer cases and around 9.7 million cancer-36 related deaths”, please replace it with newer statistics for cancer. You strongly suggested updating the whole paragraph to 2025 statistics.
Response (2) The authors have taken a keen note of the reviewer’s comment and suggestions. While we wish to agree with the call of the reviewer for 2025 statistics, we wish to share that we are restricted given the fact that the most recent estimates on the global cancer burden is presented in GLOBOCAN 2022’s report which was published in 2024- DOI: https://acsjournals.onlinelibrary.wiley.com/doi/10.3322/caac.21834 The authors posit that the GLOBOCAN 2022 report which helped to form the basis of paragraph 1 in our introduction is the most recent estimates yet had on the global cancer burden. In this regard therefore, the authors would like to retain paragraph 1 and its ensuing contents which presents the relatively recent statistics on the global cancer burden as at 2022. Kindly note that this report is cited in our list of references consulted.
See Ref #1 at line 520
Further and so as not to lose the significance of the reviewer’s observation and suggestion, the authors have inserted the following words in the introduction to enhance paragraph 1.
See lines 36-37 with projected increases occurring in several countries. For example, estimates from the United States suggest that this nation will experience more than 2 million new cancer cases and 618,120 cancer deaths in 2025 [2].
Comment (3) The COVID-19 effect should be briefly included in the abstract since it was treated in the article.
Response (3) The authors wish to thank the reviewer for pointing this out to us. To this end we have since mentioned COVID-19 effect briefly in our abstract.
See line 17 inclusive of the coronavirus disease 2019 effect,
Comment (4) In Table 1, please indicate the unit of measurement (i.e., month, year....etc).
Response (4) The authors wish to thank the reviewer for pointing out this deficiency in Table 1. The authors wish to share that this matter has since been corrected. To this end we have expanded the title of the table to read Characteristics of colorectal and prostate cancer cases in Antigua and Barbuda (2017-2021), combined and disaggregated by time-interval given in months See Table 1 at line 228
Comment (5) Tables 2 A and B are too tedious for the reader and must be summarized into one more brief table, which also applies to Tables 3 A and B. Response (5) The authors have taken note of the reviewer’s comments and have since summarized Tables 2A and B and 3A and 3B as advised. Tables 2A and 2B report on the crude estimates obtained from multinomial logistic regression analysis, while Tables 3A and 3B report on the adjusted estimates from using said regression approach. See lines 288-293 & : 363-373
|
||
|
|
||
|
|
||
|
Kindly note that in addition to the edits done in respect of the comments and/or suggestions of the Reviewer, the authors have made some edits to further improve the article and so as to ensure that there is consistency across all areas of our study re: scope and/or purpose. This included edits to text and tables. |
||
|
|
||
|
Thank you |
||

Round 2
Reviewer 1 Report
Comments and Suggestions for Authors
Proposed Title:
Timeliness of Drug Therapy in Colorectal and Prostate Cancer in Antigua and Barbuda: The Role of Disease Stage
The article addresses an important issue but contains several grammatical errors, redundancies, and awkward phrasing (e.g., "timely initiation timeliness of drug therapy"). These detract from the clarity of the message.
A thorough language and structure revision is needed to improve readability and eliminate repetition.
The topic is highly relevant to public health, especially in resource-limited settings. Investigating factors influencing timely cancer treatment is both timely and necessary.
The study’s objective could be stated more clearly and concisely, e.g.,
“This study aimed to assess whether disease stage is a predictor of time to drug therapy initiation in colorectal and prostate cancer cases diagnosed in Antigua and Barbuda between 2017 and 2021.”
The retrospective design and use of multivariable logistic regression are appropriate given the objective.
Include more details on the type of data collected and justify the use of polytomous regression, as it's not commonly used unless multiple categories of outcome exist.
The statistical findings are presented, but the explanation is somewhat unclear. Odds ratios (ORs) are reported, but their interpretation is lacking or poorly expressed.
Suggestions: Clarify the clinical implications of the OR values and explain why the lack of significance does not necessarily undermine the hypothesis.
The conclusion aligns with the results but could be expressed more precisely.
Suggested phrasing:
Disease stage was not found to be a statistically significant predictor of time to drug therapy initiation in either cancer model. However, it may hold predictive value when assessed alongside other essential patient characteristics. These findings can inform efforts to optimize cancer care protocols in Antigua and Barbuda.
Author Response
|
ROUND 2
Response to Reviewer 1 Comments
|
||
|
1. Summary |
|
|
|
Thank you very much for taking the time to review this manuscript. We do express our appreciations to you for your comments and suggestions offered. It is our hope that the revised manuscript has addressed your concerns. We do look forward to hearing from you on this. Please find the detailed responses below and the corresponding revisions/corrections highlighted in track changes in the re-submitted files.
|
||
|
2. Point-by-point response to Comments and Suggestions for Authors
|
||
|
Comment (1) Proposed Title: Timeliness of Drug Therapy in Colorectal and Prostate Cancer in Antigua and Barbuda: The Role of Disease Stage Response (1) The authors have taken a keen note of the reviewer’s comment and suggestion in respect of the title. After some discussions and understanding that the suggested topic embodies the message of the article, we have agreed to accept this suggestion. We do wish to thank the reviewer for this practical suggestion. To this end we have since made the following edits to the ‘title’ which now reads: “Timeliness of Drug Therapy in Colorectal and Prostate Cancer in Antigua and Barbuda: The Role of Disease Stage” The authors again thank the reviewer for this suggestion. Comment (2) The article addresses an important issue but contains several grammatical errors, redundancies, and awkward phrasing (e.g., "timely initiation timeliness of drug therapy"). These detract from the clarity of the message. Response (2) The authors have once more taken a keen note of the reviewer’s comments and suggestion, and notwithstanding our previously extensive language editing. We wish to assure the reviewer that we have once again subjected the article to language editing. This time around we used a different provider in MDPI Language Editing Services. See below certificate. Comment (3) The topic is highly relevant to public health, especially in resource-limited settings. Investigating factors influencing timely cancer treatment is both timely and necessary. Response (3) The authors do wish to thank the reviewer for this comment and ensuing suggestion. To this end, and after some discussion, we have since adopted the reviewer’s suggested study objective as it comports with our study’s objective and message. The authors do express our appreciations to the reviewer for sharing this suggestion with us. We thank you kindly. See lines 67-68 this study aimed to assess whether disease stage is a predictor of time to drug therapy initiation in colorectal and prostate cancer cases diagnosed in Antigua and Barbuda between 2017 and 2021. Comment (4) The retrospective design and use of multivariable logistic regression are appropriate given the objective. Response (4) The authors have taken a keen note of the reviewer’s comment and wish to share on the following: Include more details on the type of data collected: The authors having reviewed the reviewers comments have agreed to insert the following under the method’s subhead “Data collection and management”. This edit, we believe, best provides additional details on the type of data collected. See lines 88-100 We utilized a two-stage process to obtain patient data [17]. In the first phase, we used a predesigned and pretested data collection form to abstract text-based, cancer-specific, and patient-related demographic and clinicopathological information from the previously identified departments of the Sir Lester Bird Medical Centre, and electronic-based data of a similar nature from The Cancer Centre Eastern Caribbean [17]. Additional text-based data on cancer cases, including demographic, drug treatment, and socioeconomic information, were obtained from the Medical Benefits Scheme [17]. Electronic-based data on cancer deaths were collected from the Health Information Division, Ministry of Health, Antigua and Barbuda [17]. In the second phase, we cross-referenced the abstracted records using a combination of unique identifiers [17,19]. For cases collected from the Medical Benefits Scheme (MBS), we used the Medical Benefits Scheme identification number (MBS number) [19]. For records collected from the SLBMC, we utilized the hospital-generated medical patient identifier (MPI) and the MBS number. For cases from TCCEC, we relied on the unique TCCEC number and the MBS number. Concerning data on cancer deaths, we utilized the MPI and/or MBS number [19]. This approach helped to eliminate duplicate records and improve the completeness of the data collected [19].
The authors wish to thank the reviewer for urging this response. Thank you. justify the use of polytomous regression: The authors did consider using polytomous regression on the basis that our outcome variable did present us with sufficient information to allow for three distinct categories. Additionally, this form of regression analysis was chosen since it allowed us to maximize use of the available data by enabling us to consider all possible categories that existed, and without any loss of important data. It further allowed us to consider, with some ease of understanding and handling, our other covariates as potential confounders in our regression modelling analyses. Notwithstanding a point to this effect being in our limitations and so as not to lose the impact of this point in our methods section, the authors have now included a few words to justify our use of polytomous regression in the said section under subhead ‘Data Analysis’. The authors wish to thank the reviewer for urging this response. Thank you. See lines 152-156 Polytomous (multinomial) logistic regression was performed using the main outcome against the primary exposure to derive crude odds ratios (ORs) and corresponding 95% confidence intervals (CIs). Its selection was made given its advantage of effectively handling outcome variables with two or more categories [24]. Additionally, its utilization was premised on its ability to preserve essential data information, thus making it easier to manage our data and lending to a better interpretation of the eventual study results [24]. Comment (5) The statistical findings are presented, but the explanation is somewhat unclear. Odds ratios (ORs) are reported, but their interpretation is lacking or poorly expressed. Response (5) The authors have taken a keen note of the reviewer’s comments and ensuing suggestions. To this end we have made the following edit to the discussion section of the manuscript so as to accurately clarify the clinical implication of both OR values and explain the lack of significance. See lines 399-408 The odds ratios (ORs) derived from our final models point to the absence of statistical significance between the disease stage and the time to drug therapy initiation. This issue probably stemmed from our choice of sample sizes and the within-sample data variability [36]. Notwithstanding this and being cognizant of its possible effect on our hypothesis, however, the study’s findings still hold considerable clinical relevance when viewed in the context of the country’s socio-cultural experiences, existing challenges within the healthcare system, the cancer treatment models currently in use, and the ongoing epidemiological and economic burdens of the cancers studied [17,37,38]. This highlights the need for timely decisions regarding both the choice and appropriateness of drug treatment regimens linked to the management of colorectal and prostate cancers. For cancer care to be optimized, conscientious efforts, such as developing standardized colorectal and prostate cancer care algorithms, as well as offering training opportunities for clinicians and other healthcare personnel in the best evidence-based cancer care models, would be beneficial [17]. Comment (6) The conclusion aligns with the results but could be expressed more precisely. Response (6) The authors do wish to thank the reviewer for this comment and ensuing suggestion. To this end, and after a short discussion, we have since adopted the reviewer’s suggestion as it has lent improvement to our study’s conclusions. The authors do express our heartfelt appreciations to the reviewer for sharing this suggestion with us. We thank you kindly. See lines 453-455 In conclusion, this study demonstrated that where colorectal and prostate cancers in Antigua and Barbuda are concerned, disease stage was not found to be a statistically significant predictor of time to drug therapy initiation in either cancer model. However, it may hold predictive value when assessed alongside other essential patient characteristics. These findings can inform efforts to optimize cancer care protocols in Antigua and Barbuda.
|
||
|
Kindly note that in addition to the edits done in respect of the comments and/or suggestions of the Reviewer, the authors have made some other edits in respect of the second round of language editing had. This was done to further improve the clarity and readability of article and so as to ensure that there is consistency across all areas of our study re: scope and/or purpose. This included edits to text and tables. Additionally, and resulting from some of the edits that were warranted, our list of references have since increased marginally. |
||
|
|
||
|
Thank you |
||

Reviewer 2 Report
Comments and Suggestions for Authors
I see that the article is ready for publication.
Best wishes.
Author Response
ROUND 2
|
Response to Reviewer 2 Comments
|
|||||
|
1. Summary |
|
|
|||
|
Thank you very much for taking the time to review this manuscript. We do express our appreciations to you for your comments and suggestions offered. It is our hope that the revised manuscript has addressed your concerns. We do look forward to hearing from you on this. Please find the detailed responses below and the corresponding revisions/corrections highlighted in track changes in the re-submitted files.
|
|||||
|
2. Point-by-point response to Comments and Suggestions for Authors
|
|||||
|
Comment (1) I see that the article is ready for publication. Response (1) The authors wish to once again thank the reviewer for their work done in reviewing this article. We also wish to thank the reviewer for reposing confidence in the article’s readiness for publication. Thank you. |
|||||
|
|
|||||
|
|
|||||
|
Additionally, and consistent with a recommendation/suggestion of Reviewer 1, we did subject the manuscript to a second round of English language editing. This time around we used the MDPI English-Language editing services.
See copy of certificate below
|
|||||
|
|
|||||
